# Targeting Drug Delivery in the Elderly: Are Nanoparticles an Option for Treating Osteoporosis?

**DOI:** 10.3390/ijms22168932

**Published:** 2021-08-19

**Authors:** Gudrun C. Thurner, Johannes Haybaeck, Paul Debbage

**Affiliations:** 1Institute of Pathology, Neuropathology and Molecular Pathology, Medical University of Innsbruck, Müllerstraße 44, 6020 Innsbruck, Austria; johannes.haybaeck@i-med.ac.at; 2Diagnostic & Research Center for Molecular BioMedicine, Institute of Pathology, Medical University Graz, Neue Stiftingtalstraße 6, 8010 Graz, Austria; 3Department of Anatomy, Histology and Embryology, Medical University of Innsbruck, Müllerstraße 59, 6020 Innsbruck, Austria

**Keywords:** osteoporosis, nanoparticles, tissue-barriers, Howship’s lacuna, targeting

## Abstract

Nanoparticles bearing specific targeting groups can, in principle, accumulate exclusively at lesion sites bearing target molecules, and release therapeutic agents there. However, practical application of targeted nanoparticles in the living organism presents challenges. In particular, intravasally applied nanoparticles encounter physical and physiological barriers located in blood vessel walls, blocking passage from the blood into tissue compartments. Whereas small molecules can pass out of the blood, nanoparticles are too large and need to utilize physiological carriers enabling passage across endothelial walls. The issues associated with crossing blood-tissue barriers have limited the usefulness of nanoparticles in clinical applications. However, nanoparticles do not encounter blood-tissue barriers if their targets are directly accessible from the blood. This review focuses on osteoporosis, a disabling and common disease for which therapeutic strategies are limited. The target sites for therapeutic agents in osteoporosis are located in bone resorption pits, and these are in immediate contact with the blood. There are specific targetable biomarkers within bone resorption pits. These present nanomedicine with the opportunity to treat a major disease by use of simple nanoparticles loaded with any of several available effective therapeutics that, at present, cannot be used due to their associated side effects.

## 1. Nanoparticles as Drug-Delivery Agents

A nanoparticle (NP) is a nanomachine constructed according to certain principles that are still in the process of development. The use of targeted nanoparticles to transport drugs into lesions is a leading goal of nanomedicine. The core idea is to package several hundred drug molecules into one single nanoparticle (Figure 1), and to attach suitable targeting groups to the nanoparticle’s surface in order to steer it into a defined lesion [1]. The properties of the packaged drug molecules then no longer affect the bio-distribution, targeting and clearance of the drug. Instead, the properties engineered into the nanoparticle define a pre-programmed bio-distribution, with targeting and release of the drug (ideally exclusively) into the diseased site. Naturally evolved targeted nanoparticles, such as viruses, are of breathtaking complexity and show that the packaging/targeting principle is capable of great sophistication [2] and delivers active agents efficiently. At present, however, artificially prepared nanoparticles lack intrabatch homogeneity in size, interbatch reproducibility, in vivo stability, long-term storage stability and upscalability to kilogram amounts, preventing their clinical translation and industrial production [3,4,5,6,7]. This challenge of developing medically safe and effective nanoparticles is reflected in the rather low number of nanoparticles that have already entered clinical use. To date, only about 50 nanoparticulate formulations have been approved by regulatory authorities [6,8]. However, this review of osteoporosis addresses a disease with high social and economic costs, with about 200 million patients worldwide, and causing significant morbidity and mortality in elderly people [9], a disease for which nanotechnology could offer the possibility of site-directed drug delivery. Several established therapies are available but are accompanied by side effects that are often severe (see Section 8). Treatment using suitably targeted nanoparticles could reduce the frequently occurring severe side effects that accompany current treatment options [10]. However, no nanoparticle formulation has yet been approved for treatment of osteoporosis. This mini-review delineates some possibilities of developing nanoparticles for clinical application in treating osteoporosis. The main focus of this article, therefore, lies in the importance of “targeting”. The review first discusses the biological background of successful targeting in the context of the presence of tissue barriers, then moves on to discuss the physiology of bone and respective targets for osteoporosis, and closes with a discussion of potential therapeutic approaches using nanoparticles.

## 2. Targeting Tissues In Vivo

Site-directed targeting is essential to achieve site-directed drug delivery. Present-day therapeutic approaches cannot achieve this. Previously, we discussed the many barriers that must be surmounted to achieve target access from the blood or from the gastrointestinal tract [11,12,13]. In general, peroral and intravasal routes of drug application are extremely inefficient (efficiency a fraction of 1%) and have narrow therapeutic windows due to their multiple and serious side effects [14,15,16]. Application to mucous membranes is almost equivalent to intravasal application, the difference being the passage across a thin epithelium with narrow connective tissue between epithelium and blood vessel. Orally applied agents encounter the strong acid milieu and mechanical churning in the stomach, along with exposure to highly active digestive enzymes. When the therapeutic agents finally enter the blood, from there, they must overcome blood-tissue barriers to reach their sites of action. Throughout their hazardous passage through gut environments and the blood milieu, they must retain their chemical integrity and biological activity. Agents injected directly into the blood must survive the shear forces, enzymes and immune cells in the bloodstream environment. During their hazardous journey through the gut and bloodstream, targeted nanoparticles must retain their targeting groups, typically an antibody molecule or one of its fragments. The risk is that these groups may succumb to enzymes present in gut fluids or embedded in gut epithelia, or ones present in the blood. Having arrived in the vascular bed of the organ containing the target lesion, small molecules can diffuse into the tissue, arriving with a much reduced concentration at the lesion. Nanoparticles are far larger and cannot diffuse across cell membranes or pass through intercellular junctions. Whereas small drug molecules are usually a few nanometers in size, nanoparticles have diameters exceeding 20 nm and are often as large as hundreds of nanometers. They must be carried across tissue barriers by specific uptake mechanisms before they can access and enter the target cells, and to do this, they must interact specifically with cellular uptake mechanisms. Thus, truly specific tissue-targeting from the blood requires at least two targeting groups: the first directs the particle exclusively towards the lesion-specific endothelium and aids crossing this barrier, while the second directs the nanoparticle to the target cells after diffusion through the interstitium [11]. Although such true (“vectored”) targeting offers the potential of targeting efficiencies near 90% [17], it is more difficult, elusive and expensive [18] to achieve than “enhanced permeability and retention” (EPR) [19], which relies on the tendency of nanoparticles to accumulate in lesions with increased vascular permeability [11]. Some organs (Figure 2) have raised vascular permeability under physiological conditions in the healthy state and accumulate nanoparticles; these include, for example, the liver, spleen, kidneys, bone marrow, lymph nodes, pancreas, choroid plexus and the pituitary gland. Such accumulations have long been documented for metal nanoparticles, including silver and thorium [20], gold, copper, zinc, technetium, aluminum and iron. If these organs accumulate blood-borne nanoparticles that are applied with the intention of treating diseased tissues and bearing potent drugs, the nanoparticles might deposit unwanted high drug concentrations in one or more of these organs. Drug-loaded nanoparticles accumulating by EPR thus carry the risk of significant side-effects. It is evident that truly specific targeting by use of orally or intravasally applied nanoparticles involves considerable technical difficulty and expense, followed by further difficulty and expense during licensing for clinical use. To achieve true targeting, it is essential to understand the functioning and the physiological regulation of the body’s tissue barriers, in particular, those of the blood- and lymph-vascular system (Figure 3). Similar problems occur when attempting to use EPR for “targeting”.

The routes of application of common anti-osteoporotic agents such as, e.g., bisphosphonates or selective estrogen receptor modulators (SERMs) comprise oral, nasal and systemic pathways [21] and thus are burdened with the barriers and side-effects just mentioned. Many of these difficulties fall away, however, if the lesion is located in the blood or borders directly on the blood milieu. This is one reason why the earliest chemotherapeutic successes were with leukemias, which are cancers arising from blood cells; successes with “solid” tumors followed rather later. This is significant for the remainder of this review: truly specific targeting to a lesion site is greatly simplified when the target cells are directly accessible from the compartment into which nanoparticles are applied. For example, in the case of the bloodstream, atherosclerotic lesions are in contact with the blood and can be targeted by a single targeting group, such as an antibody specific for Apo-E. As a further example, in the case of oral application, stomach ulcers are in direct contact with the gastric lumen and are, therefore, easily accessed by drug molecules or by nanoparticles. As such, a major consideration for targeting in vivo is that in local application, with direct access to the target molecule, a single targeting group suffices to direct the nanoparticle to cells in the lesion of interest.

In this review, we present evidence that osteoporosis represents a rare opportunity to attempt nanoparticle therapy of a major disease by use of rather simple NPs. Identification of a good target and a suitable drug will depend as much on the biology of osteoporosis as on the design of the nanoparticle.

## 3. Regulation of Blood-Tissue Barriers: Four Levels of Permeability

Nanoparticles do not generally cross blood-tissue barriers because they cannot cross impermeable endothelial cell layers. However, endothelial barriers vary widely in permeability. Vascular permeability factor (VPF)/vascular endothelial growth factor (VEGF), a cytokine which uniquely combines survival-promoting, permeabilizing and angiogenetic effects on endothelial cells [22,23], is the major regulator of permeability. In both normal and diseased tissues, cells may store large amounts of VEGF, but do not secrete it into the surrounding interstitial compartment [24]. They secrete it in response to hypoxia [25] and to numerous cytokines, hormones and growth factors. Disease states are characterized by high VEGF concentrations (>50 pmol/L), which initiate neo-angiogenesis [22], the formation of new microvessels. During the early steps in neoangiogenesis, the endothelial cells hydrolyze the subendothelial basement membrane, then detach from it and retract from one another, leaving gaps in the endothelial cell layer [26], thus abolishing all barrier functions. This pathological permeability is exploited by EPR. In contrast, in healthy tissues, VEGF is used to exert a finely dosed control. At low concentrations (<0.4 pmol/L) [27], it activates intracellular signaling in endothelial cells, and the sub pmol/L concentrations of it produced by pericytes and other tissue-specific epithelial cells [28,29] serve throughout the vasculature to support endothelial survival [30]. An example of this **first, lowest level of permeability**, is the blood-brain barrier, allowing only a very restricted range of molecules into the central nervous tissue (CNS), and thus providing an unusually protected milieu for neuronal function [31]. The **second level of permeability** is found in most other tissues, which exchange nutrients and metabolites between the blood and the tissues via capillaries; we reviewed these first two levels of permeability previously [11,12,32]. Small molecules and a range of macromolecules can pass them, but nanoparticles and cells do not. A **third, significantly higher level of permeability** is found in organs whose function requires rapid and easy transfer of large volumes of small molecules, macromolecules, nanoparticles and even cells between the blood and the organ’s interstitial space (Figure 3). Here, VEGF at concentrations of 2–5 pmol/L maintains a specialized endothelial differentiation state in permeable sinusoidal microvessels [23,33]. These are highly, but not totally, permeable: they maintain a “low” barrier function, which can be abolished by irradiation [34]. They are found in the spleen and in the renal glomeruli [29], the gastrointestinal tract, the female reproductive system, the circumventricular organs of the central nervous system [35,36], the endocrine organs [29] and the bone marrow [37] (Figure 2). To achieve this relatively high level of vascular permeability, VEGF acts by several mechanisms [22]. It attenuates endothelial cells and causes the enhanced appearance of pores and fenestrae [35], vesiculo-tubular structures [38] and vesiculo-vacuolar organelles/caveolae [23], and it influences the integrity and function of all three types of inter-endothelial junction. In particular, it loosens the adherens junctions, which connect adjacent endothelial cells to form continuous sheets of cells; such a loosened endothelium permits free passage of entire cells [26,39]. Sinusoidal endothelial cells lack a complete basement membrane and are actively phagocytic [40]; they filter out both natural and therapeutically applied nanoparticles from the blood [20], accumulating them within lysosomes (Figure 4). During aging [41], and in certain diseases (e.g., liver cirrhosis and fibrosis) [42], the sinusoidal microvessels may revert to the more usual capillary form (“pseudo-capillarization”), losing their regulated state of raised permeability. Finally, there are rare but important exceptions to the general rule that nanoparticles cannot cross blood-tissue barriers and extravasate into the tissues. At sites in sinusoidal capillaries where constitutive VEGF expression by local cells such as hepatocytes [29,33] leads to formation of pores in the endothelial cell walls, the **fourth level of permeability** removes all barriers to nanoparticles, allowing them direct access to the tissue interstitial space. This is of special importance in nanomedicine. The liver, with its sieve-like arrangement of endothelial cell pores lacking diaphragms, is the best example. Although the liver sinusoidal endothelium is often described as “fenestrated”, it in fact has diaphragm-free pores that form open connections between the lumen of the sinusoid and the Disse space [43], the interstitial compartment of the liver, and allow free passage of nanoparticles smaller than a certain size. The liver sieves natural nanoparticles (chylomicrons) [44,45], a function of great importance in connection with the artificial nanoparticles being developed for use in nanomedicine. Its sieve area is large, approximately 4.5 m^2^ of the sinusoidal capillary surface (an adult human has ~1.5 m^2^ of skin surface) [44]. Lymph nodes provide a further example as they have both a blood and lymphatic circulation; the lymph capillaries are sinusoidal and possess pores. Another example, of major importance in this mini-review, is found in the bone.

## 4. The Micro-Anatomy of the Howship’s Lacuna

The dynamic remodeling of bone occurs via a finely regulated set of interactions that involve the interplay of two major cell types on the surface of bone. The phagic osteoclasts first demineralize and then partially digest the organic material comprising the non-mineral bone substance. The bone being remodeled is laid down by the second cell type, the active osteoblasts, which play a role in regulating the activity of the osteoclasts. The interplay between these two cell types on the surface of remodeling bone occurs at the nexus between endocrinological and homeostatic systems. It has been examined in a range of species and for several varying types of bone, and also in comparison with similar mechanisms that occur in teeth. The sequence begins when a mesodermal cell, the future osteoclast, settles on the surface of bone and flattens to form a circular cell attached at all points of its perimeter to the bone surface. This cell is only a few micrometers (µm) in size, and between its cell body and the bone, it forms a shallow compartment, into which it releases protons, rendering the compartment acid and dissolving the bone mineral. The acid-eroded bone surface is penetrated by the stumps and remains of collagen fibers that previously formed bundled scaffolding within the bone mineral. The lysozymal secretions of the osteoclast digest away part of all of these organ constituents. The osteoclast can migrate, moving away from its original eroded and digested surface, with bursts of speed in excess of 10 µm/h. In this way, it leaves a track that can be circular, meandering or straight; the osteoclast does not follow marks or grooves in the bone surface. The osteoclast matures into a much larger cell, which is syncytial: it soon contains a few cell nuclei and as it grows, the number of nuclei can become very large, for example, the cell may have 50 nuclei. The surface area beneath the mature flattened osteoclast increases and can become as large as hundreds of µm^2^. This bone area beneath the osteoclast is subjected to a process of demineralization and partial organic removal that is termed “resorption”. The resorbed area of bone has received numerous names, including “bowl”, “track”, “area”, “patch”, “lacuna”, “cavitation”, “bay”, “tongue” and others. A word of caution: the word “lacuna” is used to mean several quite different things in bone and tooth terminologies, including the general bone resorption area, the Howship’s pit itself, or the cavity in which the mature osteocyte resides, far away from any resorption. Osteoclasts may occur singly or in groups, may move away from one another to form patterns resembling the “fairy rings” formed by fungi in woodlands, or may simply spread by growth. The resorbed region may be hundreds of µm^2^ in size. At certain points within these areas, the osteoclasts may penetrate deeply into the bone, forming pits that are known as “Howship’s pits” (Figure 5). These may be as many as 25 µm deep and have steep sides, with entrances that are several µm across, or they may be flatter and take the form of shallower depressions, with wider entrances and depths of only several µm. The depth of penetration depends on the degree of mineralization encountered by the probing osteoclast [46]. The sequence described above is a summary of the findings reported in an extensive series of papers over a 20-year period (about 1954–1985), resulting largely from work by the research team associated with Prof. Alan Boyde. Prominent amongst the series of papers were these: [46,47,48,49,50,51,52,53,54,55,56,57,58,59,60]. A smaller number of authors have considered the distribution of resorption areas throughout the skeleton: for example, in rabbits [61] or human [62]. For the application of nanoparticles, the size and shape of resorption pits are significant parameters. Narrow and deep pits—for example, 6 µm wide and 20 µm deep—are unlikely to have strong flows of blood circulating in them and might be best targeted by use of the smallest size nanoparticles. The measurement of Howship’s pit depths is technically difficult and has required ingenuity and specialized instrumentation to achieve reliable measurements [58,59,60]. The pits often reach volumes of 1000 µm^3^, and volumes of 2000 µm^3^ are not rare; much more rarely, volumes as large as 10,000 µm^3^ can be found. A pit of width 10–40 µm and depth 4–6 µm can be considered “normal” [60]. As is often the case in nanomedicine, the biological conditions within the target tissue must be considered equally as carefully as the design of the nanoparticles to be employed.

## 5. Regulation of Bone Homeostasis

In the bone marrow, sinusoidal microvessels maintain a blood-tissue barrier comparable to that in the spleen [37], at “level 3” permeability. However, in the tiny pits comprising the Howship’s lacunae where bone remodeling occurs, osteoblasts constitutively secrete large amounts of VEGF [63]. This creates a miniscule barrier-free “vascular compartment” [64] (“level 4” permeability equals total permeability) around the microvessel in each pit (“Howship’s lacuna”) where bone is being remodeled (Figure 6). Bone is highly vascularized [65], its complex anastomosing network of arterioles and sinusoidal capillaries supplying materials and cells to the BMUs (basic multicellular units) located in Howship’s lacunae. Ossification originally occurred in close proximity to vascular ingrowth, and tightly regulated vascular invasion continues as the BMU migrates. Each BMU contains a sinusoidal microvessel lying within 100 µm of the osteoclasts dissolving bone and the osteoblasts replacing it [64]; the vessel recruits cells for bone repair [65]. As the BMU migrates across the bone, its resident microvessel grows in length, stimulated by the high concentrations of VEGF, which also keep the sinusoid endothelia totally permeable. Although numerous internal signaling interactions maintain the BMU as a functional unit, the major regulation of BMU activity in skeletal homeostasis is hormonal [66]. Besides its stabilizing role, bone can be regarded as an endocrine organ, secreting osteocalcin into the blood [67]. The Howship’s lacuna is at the hub of several interlinked regulation cycles, including the ones regulating energy metabolism in the insulin cycle [68]. Important hormones at the BMU include PTH and calcitonin, but the key player in hormonal regulation of BMU function is estrogen. Osteoblasts [69], osteoclasts [70] and the local endothelial cells [71] all bear estrogen receptors. Estrogen protects osteoblasts against apoptosis and stimulates their osteogenic activity [72]. Osteoblasts express receptor activator of NF-κB ligand (RANKL) [73], which induces maturation of osteoclasts [74], but they also secrete a soluble decoy receptor, osteoprotegerin (OPG), which neutralizes RANKL [75]. Estrogen suppresses osteoblasts’ RANKL production [76] and increases OPG production [77], and thus acts on the RANK-RANKL-OPG axis [78] to reduce osteoclast activity. In addition, estrogen inhibits osteoclast formation from mononuclear hematopoietic stem cells. In sum, therefore, an estrogen deficiency increases the RANKL/OPG ratio, strongly reducing bone-building activity but allowing bone resorption by osteoclasts to continue [79].

## 6. BMU Dysregulation: Primary Osteoporosis

The permanent reduction in estrogen production after the menopause leads to primary osteoporosis, a systemic atrophy of bone that attenuates bone micro-architecture [80]. The hormonal changes of the menopause do not affect the “constitutive” remodeling of bone: osteocytes continue to respond to bone strain and to micro-fractures by initiating bone remodeling. However, the osteoclast-regulating influence of estrogen is strongly reduced, so the two functions of the BMU (bone resorption, bone replacement) are out of balance or are uncoupled [81] and resorption occurs faster than bone replacement [82,83]. The difference between bone calcium accretion and loss almost doubles in postmenopausal women [82]. After perforation and ablation of trabeculae has occurred, bone replacement is eliminated and the attenuation becomes permanent [84]. Most bone is lost during the first 3–6 years after menopause [85], but loss related to low estrogen levels may continue for longer than 20 years [86]. In primary osteoporosis, which has a histology similar to geriatric bone, the sinusoid density in a 50-year-old woman is almost identical with that of a 70-year-old woman; the mass of trabecular bone is correspondingly reduced. Osteoporosis is recognized as a major health problem with a large socioeconomic burden [87,88].

## 7. Possible Targets for Nanomedicine in Osteoporosis

Primary osteoporosis is characterized by withdrawal of estrogen from the bone-repair team of osteoblasts and osteoclasts in the Howship’s lacuna. The lacuna has three physical characteristics, which can be exploited to deliver drugs to it. First, nanoparticles have full access to the bone-repair- and bone-resorptive cells directly from the blood. Second, the mineral component is uniquely accessible to nanoparticles in the bloodstream. Third, the low pH milieu provides a means of selectively uncoupling the drugs from the nanoparticle, precisely at the point of desired action by pH-sensitive drug linkages. Also, the presence of lysozymal enzymes can disintegrate organic nanoparticles. In addition to the assessment of bone mineral density (BMD) by dual-energy X-ray absorptiometry (DXA), bone turnover markers (BTM) have become increasingly important for diagnosis and monitoring of disease progression. They are divided into three categories: (1) bone formation markers, (2) bone resorption markers and 3) regulators of bone turnover. In addition to these markers, proinflammatory cytokines such as, e.g., IL-1β, IL-4, IL-6, IL-15, IL-17 and TNFα play a role in osteoporotic degeneration of bone and cartilage; however, their role as biomarkers is not yet explicitly identified [89]. Table 1 gives a summary of presently available BTMs and their clinical utility in the assessment of therapy and disease progression in osteoporosis; for deeper study of these biomarkers, see, for example, the extensive reviews of [9,90,91,92]. Despite their potential clinical use as biomarkers in the assessment of osteoporosis, several of these molecules present in the lacuna additionally offer chemically specific targets for drug delivery. We consider the following targets with a combination of features that should be exploitable to allow appropriately designed nanoparticles to accumulate in the bone-repair site (by a mixture of EPR and true targeting), releasing a drug there in a milieu-specific fashion:

**Bone Mineral (Hydroxyapatite)**: Bisphosphonates have already been studied for use in directing drug molecules and nanoparticles to bone [93,94]. The results are mixed, and it is evident that appropriate linker-bisphosphonate combinations will need to be identified.

**RANK, RANKL, Osteoprotegerin**: This triad, offering a prime set of targets in bone diseases including osteoporosis, is produced by a range of cells belonging to the functionally interlinked osteoarticular, immune and vascular systems [78]. Anti-RANKL antibodies such as denosumab are already in clinical use in osteoporosis therapy, however, with sometimes severe side-effects requiring hospitalization [10,78,95]. OPG and soluble RANK can also be delivered by use of adenoviral vectors [96]. It has been suggested [78] that polymer delivery systems might replace this rather toxic form of application, a suggestion that fits well with the options discussed in this review. 

**Calcitonin**: This powerful inhibitor of osteoclast activity [97] is already in use in osteoporosis therapy, though the doses used are much in excess of the endogenous rate of production [98]. Approximately 70% of patients react by generating neutralizing antibodies to the salmon calcitonin used, and this immune response can lead to secondary resistance to treatment [99]; targeted nanoparticle-based therapy could avoid this response.

**Osteocalcin (or bone γ-carboxyglutamic acid protein, or bone Gla protein)**: This protein hormone, being part of the non-collagenous bone matrix, is synthesized by osteoblasts and odontoblasts [100]. γ-carboxylated osteocalcin binds to the calcium of bone mineral, and in doing so, inhibits excessive bone mineralization. Osteocalcin might even play a role in lowering the blood glucose level directly, by stimulating insulin synthesis, and indirectly, due to stimulation of adiponectin; however, these findings still need to be verified [101]. As osteocalcin is a specific bone mineral protein, it represents a potential target with high sensitivity and specificity. Antibodies against osteocalcin are available commercially.

## 8. Possible Drugs for Nanomedicine in Osteoporosis

Numerous drugs have been developed and evaluated in the context of osteoporosis. The drugs in clinical use are mainly divided into two groups: (1) anti-resorptive drugs such as bisphosphonates, SERMs (e.g., raloxifene, bazedoxifene and tamoxifene), denosumab or calcitonin, which inhibit osteoclast activity, thereby exerting influence on bone mass; (2) bone-anabolic drugs such as rPTH, estrogen and teriparatide [102], abaloparatide [103] or romosozumab (a sclerostin antibody) [104], which induce bone formation by exerting an influence on bone-remodeling [10]. Each of the mentioned drugs has the potential to treat osteoporosis effectively, but each is also presently restricted in its use due to potentially severe side effects [10]. As examples: **Bisphosphonates**, having been discovered 50 years ago, have been in use as anti-resorptive agents for nearly 30 years [105]. They strongly inhibit osteoclast activity, leading to uncontrolled bone growth [106]. Yet, side effects such as atypical femur fractures [107] or osteonecrosis of the jaw limit their long-term use in osteoporosis treatment [108]. **Estrogen:** In 1992, Riggs & Melton [83] recommended estrogen as the mainstay of both prevention and treatment in post-menopausal women. Estrogen applied as hormone replacement therapy (HRT) is an effective therapy for primary osteoporosis. Estrogen/progesterone combinations as hormone replacement therapy retard bone resorption and provide significant increases in bone mineral density with minimal withdrawal bleeding and significantly reduced rates of osteoporosis. If begun soon after menopause, estrogen therapy prevents the early phase of bone loss and decreases the incidence of osteoporosis-related fractures by about 50%. In women with established osteoporosis, estrogen increases the mean vertebral bone mass by more than 5%, decreases the bone turnover rate and reduces the vertebral fracture rate by half [109]. However, non-skeletal deposition leads to several side effects, so compliance is poor [110]. These side effects are: (1) monthly withdrawal bleeds to prevent uterine cancer, (2) increased risk of breast cancer due to the estrogen stimulating any malignant cells possessing estrogen receptors, (3) weight gain and breast tenderness and (4) an increased rate of endometrial carcinoma. Treatment with estrogen may frequently be required for longer than 20 years [83]. Present protocols for treating metastatic breast cancer abolish all estrogen from the body. The treatment protocols include anti-osteoclast agents, but in the absence of the major bone-building driver estrogen, these permissive agents will not rebalance bone dynamics—osteoporosis is, therefore, associated with treatment for breast cancer metastasis. Due to the side effects and increased oncological risk, estrogen is no longer a standard treatment for osteoporosis [83]. It is interesting, for this review, that estrogen replacement therapy has side effects due to the interaction of estrogen with non-bone cells at non-skeletal sites. **BMP-2**: This peptide has become an important tool in healing bone defects [111]. The side effects resulting from use of BMP-2 such as tumorigenesis and uncontrolled bone formation are due to its action at inappropriate sites [112], and could be eliminated if BMP-2 could be directed via a targeting procedure to its desired site of action. These are only a few examples of effective anti-osteoporotic drugs, and these have limited use due to their severe side effects. Limiting these side effects by site-directed drug application is the major driver of nanoparticle research focusing on osteoporosis. This is also reflected by the immense and active literature that is available [7,113,114,115,116], and nanoparticles for such applications are under continuous development [94,111,117,118,119].

## 9. Therapeutic Approaches Using Nanoparticles

For nanoparticles, the route of application presupposes certain important design criteria for the size, charge, matrix material and targeting moieties, e.g.,:local application via implants or bone marrow injections would not require a targeting moietysystemic application via intravenous injection would require only one targeting moiety, aimed at a target found exclusively in the bone resorption lacunaeoral application would require at least two targeting moieties, together with protection against the gastric milieutransdermal application would also require at least two targeting moieties, and in view of treating osteoporosis, is only reasonably possible for small-sized nanoparticles; on intact skin, such nanoparticles would additionally require cell-penetrating peptides so that the nanoparticles could reach the vascularized structures of the deeper skin layers; otherwise, the skin needs to be disrupted by artificial means [120].

In osteoporosis, pathogenesis proceeds at one of the few sites that have no blood-tissue barrier. As a result, only a single type of targeting group needs to be attached to a nanoparticle to treat this disease. We, therefore, consider osteoporosis an opportune model disease for nanomedicine since its biology renders it ideally suited to the development of solely targeted drug-bearing nanoparticles for therapeutic purposes. The pathogenic mechanism involves a disbalance in the functioning of the BMU, and therapy will ideally redress the internal functional balance of the BMU. The packaging and targeting options, which are the characteristic advantages of nanoparticles, can be used to enhance the targeting efficiency and overall therapeutic efficacy of agents influencing the BMU functional balance. The enhanced targeting capabilities built into nanoparticles should reduce the deposition in non-skeletal tissues, thereby minimizing the side effects. In principle, any of the known organic and inorganic nanoparticle types can be developed into a bone-targeting device. So far, the only clinical study into treating osteoporosis with nano-based materials used organic nanoparticles [118]. Qu et al. compared pluronic nanoparticles (PG) with “oligosaccharide nanomedicine of alginate sodium” (ONAS) particles. Both nanoparticle types were mixed with ampicillin and administered as daily oral dosages. The study included 96 patients suffering degenerative lumbar disease; the group was divided evenly, with 48 patients receiving PG and 48 patients receiving ONAS. It is indicated that the results favor ONAS over PG nanoparticles with respect to side effects, infection and fusion rates; however, we await confirmation. Table 2 gives a brief overview of recent developments in using nanoparticles designed for treating osteoporosis:

## 10. A Suggestion for a Nanoparticle Targeted to the Howship’s Lacuna

Bone resorption lacunae offer highly distinctive targets. The osteoclasts migrate away from surfaces that they have demineralized, leaving fields of collagen stubble behind them [46,56]. Free collagen stubs also protrude from the walls of the Howship’s lacunae. The collagen stubble is later cleared away by the bone-lining cells [135], so its presence signals an active bone resorption site. There is only one other location in the bloodstream at which collagen can be encountered, and there, only in vanishingly small amounts: this is the Disse space of the liver. Active bone resorption sites signal their presence with highly distinctive fields of collagen stubble, and can, therefore, be targeted by attaching anti-collagen I antibodies to nanoparticles. Active osteoclasts secrete lytic enzymes such as cathepsin K into the compartment beneath them, so that the active bone resorption field contains slightly raised concentrations of these enzymes. A protein nanoparticle docked to the collagen stubble is exposed to these enzymes, and can, therefore, decompose in a process taking several hours. During nanoparticle decomposition, drug molecules integrated into the nanoparticle protein matrix will be set free into the active resorption region. The movement of blood across the resorption patch would prevent these drug molecules reaching any significant concentration there. However, the ~1000 µm^3^ volume within the Howship’s pits is unlikely to be swept clear by fluid flow, and this offers the possibility that a drug released from the dissolving nanoparticle could reach locally raised concentrations within the most active bone resorption space, the Howship’s pit. Osteoclasts are reportedly highly sensitive to calcitonin [97,98] or cytochalasin B [61], and it is also possible that their proton secretion could be reduced by proton pump inhibitors. It is of some importance that side effects from these three types of drug just mentioned would be mild: these molecules are unlikely to concentrate at any sites other than active bone resorption sites, or possibly in the Disse space of the liver, which is well capable of metabolizing them. In sum, bone resorption lacunae offer distinctive targets to which drug-loaded protein nanoparticles can be docked with high targeting specificity by use of readily available and inexpensive targeting groups, and at which the slow dissolution of those nanoparticles can release highly active drug molecules over many hours. Suitably dosed continuous application of such nanoparticles in the bloodstream could subdue osteoclastic activity and significantly ameliorate the condition of osteoporotic individuals.

## 11. Conclusions

In healthy individuals, very few tissue sites are entirely free of blood-tissue barriers, and such sites allow nanoparticles injected into the bloodstream to have direct and immediate access to them. Only one targeting group is needed to steer the nanoparticles to these sites, reducing the dosage and accumulation in all other tissues, and expediting clinical development and licensing. One such site is the bone remodeling site, within the bone resorption pits. A major disease of the elderly, osteoporosis, is caused by physiological and iatrogenically caused imbalances at this site. Several therapies for osteoporosis have proven to be effective, but they encounter poor patient compliance and thus become ineffective, due to side-effects caused by non-skeletal deposition of the therapeutic agents. Osteoporosis offers a uniquely “easy” target for a nanomedical approach to a major disease, via nanoparticles bearing any one of several effective drugs, solely targeted to specific chemical features present uniquely in the bone resorption pits. The possibility of using such nanoparticles to avoid non-skeletal deposition, and hence to strongly reduce or eliminate side-effects such as triggering of cancer or enhancement of cancer progression, motivates a rethink about which therapeutic approach to osteoporosis offers most benefits to the increasingly large number of women at risk. This mini-review has discussed intravenously applied nanoparticles, but development of orally administrable nanoparticles would result in much higher compliance, especially in older people. Thus, further studies are needed to address this highly relevant clinical issue.

## Figures and Tables

**Figure 1 ijms-22-08932-f001:**
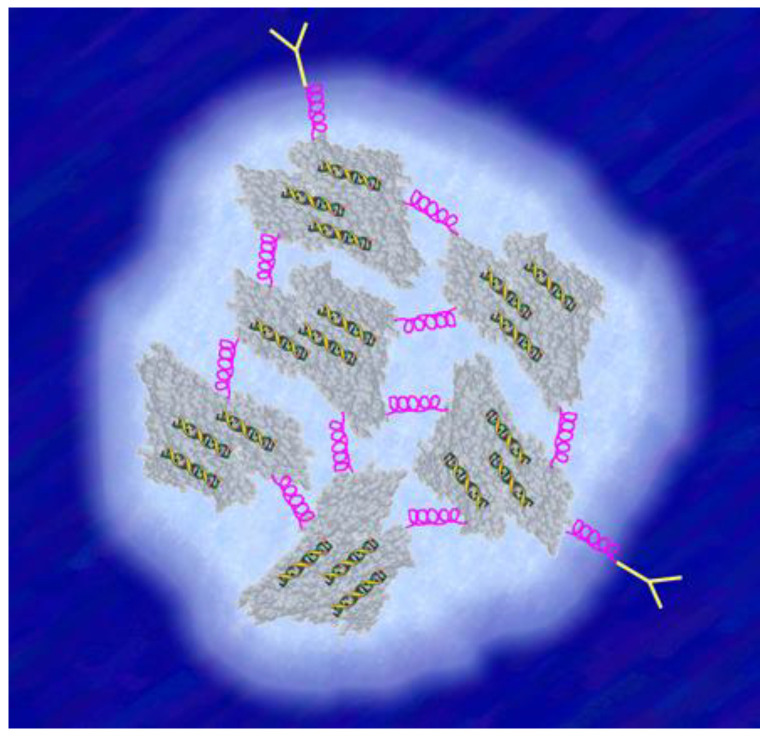
A targeted, drug-bearing albumin-based nanoparticle: the drug (siRNA; yellow-green spirals) is incorporated into the albumin molecules, enabling albumin to act according to its nature as the body’s main carrier for endogenous and exogenous substances in the blood. Each albumin is connected via stable but still flexible linker molecules (pink spirals) to give the nanoparticle a thixotropic character important for transmembrane passage. The yellow Y-shaped molecules depict antibodies important for targeting.

**Figure 2 ijms-22-08932-f002:**
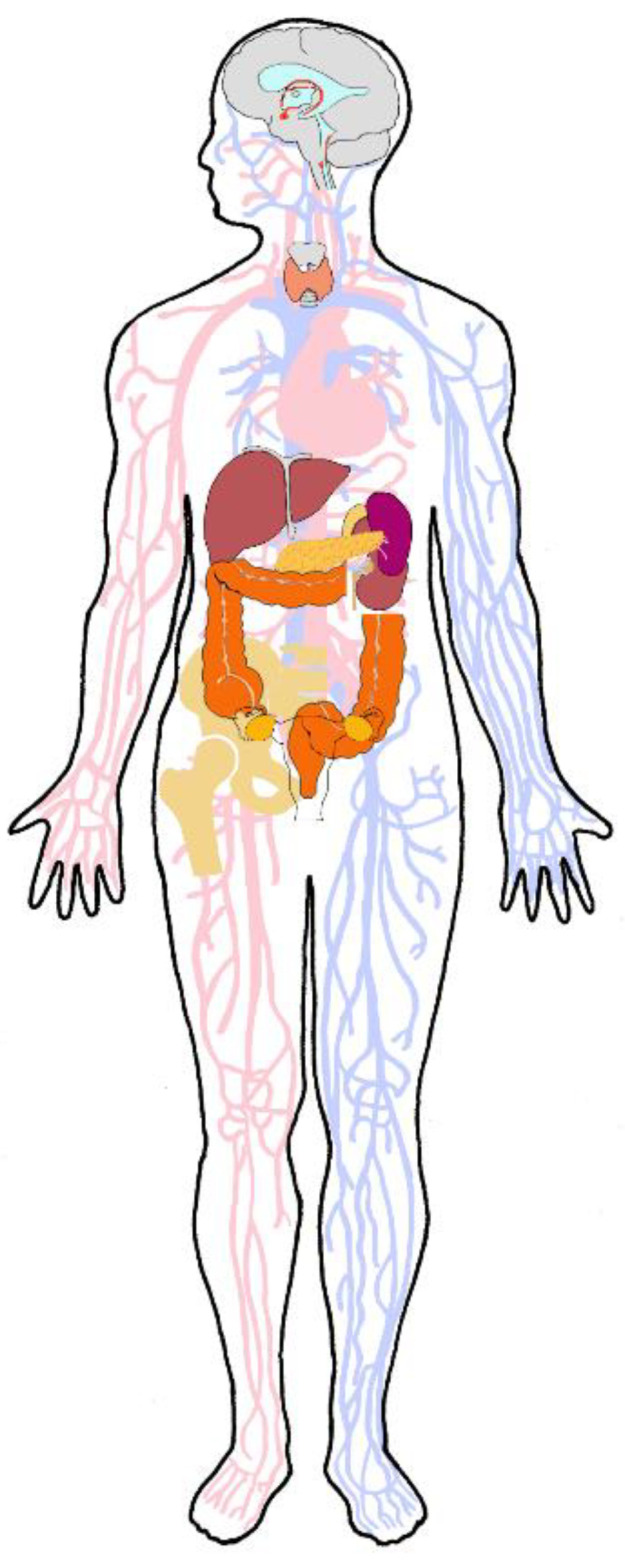
Sketch of the human body, highlighting the organs containing sinusoidal microvessels, which are, therefore, likely to accumulate nanoparticles by endothelial phagocytosis and by the EPR effect; these sinusoidal microvessels exhibit the vascular permeability levels 3 and 4 noted in the main text. During acute exposure to blood-borne nanoparticles, the liver fills with them rapidly, and bone resorption pits likewise rapidly accumulate them. Higher doses—and chronic repeated application of nanoparticles—also lead to accumulations of nanoparticles within the cells comprising the sinusoidal microvessels in these organs (liver, kidney cortices, spleen, gut, ovaries, bone resorption pits, endocrine organs (thyroid and adrenal glands, pancreatic islets, ovaries) and the circumventricular organs of the brain). Accumulation of nanoparticles at these sites is a potentially dangerous likelihood during chronic/repeated application of nanoparticles.

**Figure 3 ijms-22-08932-f003:**
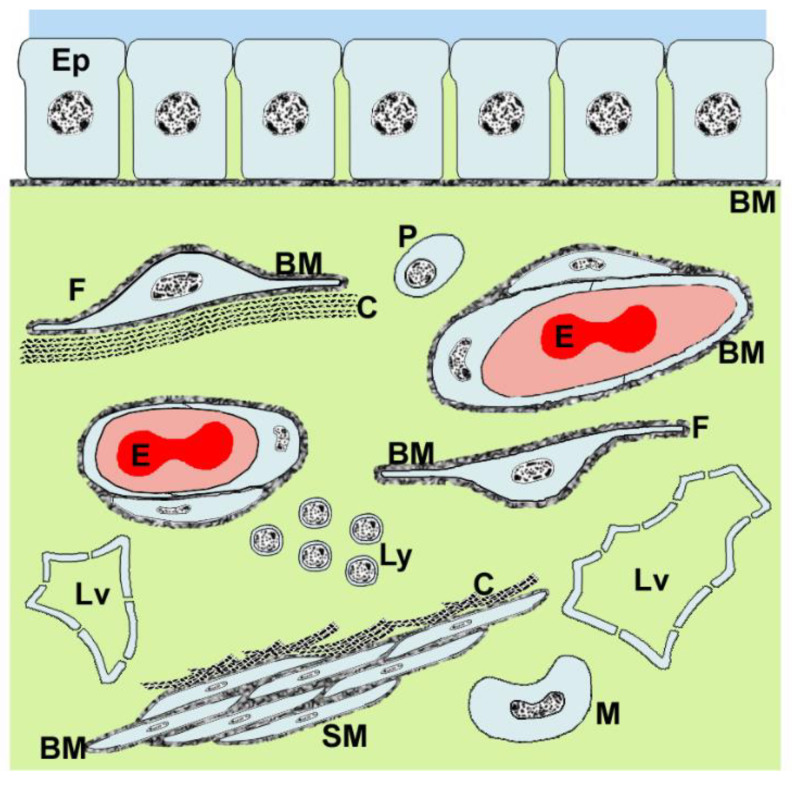
Generalized sketch highlighting the four liquid compartments that are present in all tissues. Epithelial cells (Eps) interconnected by junctional complexes divide two aqueous compartments, namely, the apical compartment (bright blue) and the interstitial compartment (yellow-green). The interstitial compartment is emphasized in this sketch, which for clarity, omits the complex fibrillar structures usually present. The concentrations of free cytokines and growth factors in the interstitial compartment are of critical importance because they directly influence cell behaviors within this compartment; they are, however, technically difficult to measure in most tissues. The interstitial compartment contains a range of cell types, some shown here (F: fibroblast; P: plasma cell; Ly: lymphocytes; SM: smooth muscle cells; M: macrophages). In addition, it contains the open endings of the lymphatic vessels (Lvs), into which the interstitial fluid flows and is then termed “lymph”. The microvessels (containing blood and erythrocytes (E) in this image) contain the third liquid compartment (blood). The fourth liquid compartment is the cytosol, present in each of the cells and of a specific composition within each cell type, shown here as pale blue. Basement membranes (BMs) are shown to be associated with several of the structures, including the epithelium, the endothelium and cell types such as fibroblasts and smooth muscle cells. Collagen bundles (C) are shown associated with fibroblasts and smooth muscle cells. A typical endothelial-epithelial barrier consists of the endothelial cell layer with its intercellular junctions and its basement membrane, the intervening extracellular matrix containing fibers and cells, the subepithelial basement and then finally, the epithelial cell layer with its intercellular junctions. Scale of the sketch: the capillary blood vessels shown here are usually 5–7 µm in diameter.

**Figure 4 ijms-22-08932-f004:**
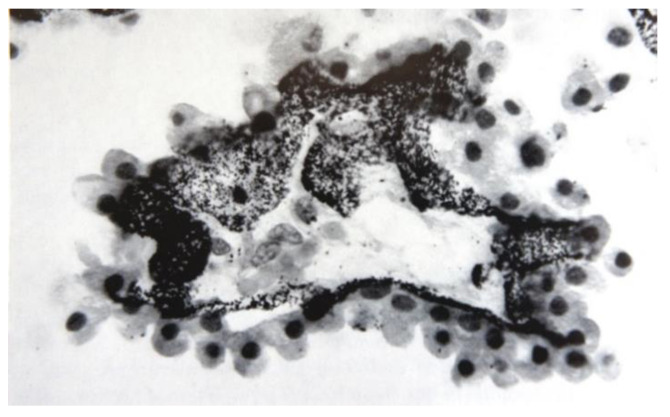
Human choroid plexus; post-mortem, the endothelial cells are packed with phagocytosed silver nanoparticles, which have been applied as an oral anti-ulcer medication (“Targesin^®^”) over the course of several years. The nanoparticles in this colloidal formulation have crossed from the gut lumen into the blood, and from there, been taken up by the choroid plexus endothelial cells, where they accumulate. Chronic application of drug-loaded nanoparticles could result in high exposure of the choroid plexus to potent agents; similar considerations apply to all the organs highlighted in Figure 2. This illustration is copied from [20], with kind permission of the author. Original magnification: 400×, image size: 10.3 cm wide, 6.8 cm high.

**Figure 5 ijms-22-08932-f005:**
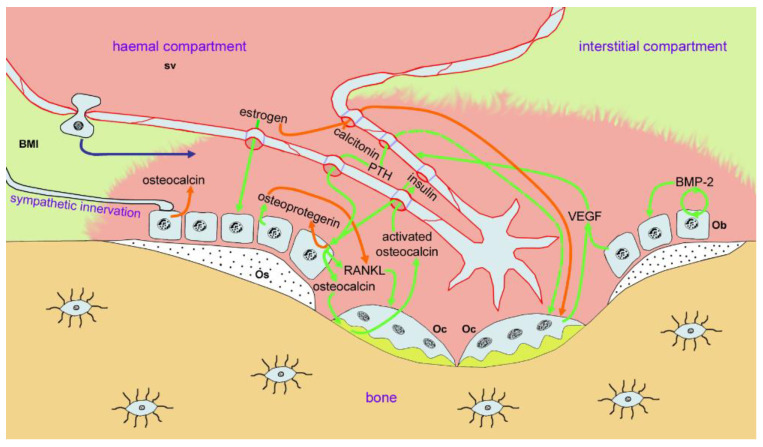
Schematic of a bone resorption pit (“Howship’s lacuna”). Two (multinucleated) osteoclasts (Ocs) are shown sealed to the mineral of the bone trabecula and using acid secretions to dissolve the mineral (bright yellow compartment), thus forming a pit or trench in the bone material. Close to the osteoclasts, several osteoblasts (Obs) secrete osteoid (Os, stippled) to initiate replenishment of the bone material. A neo-angiogenetic (sprouting) microvessel, extending into the pit from the sinusoidal vessel (Sv) within the bone marrow interstitium (BMI; pale green), is totally permeable (indicated by sketched pores (blue) in the endothelial walls), and therefore, allows the full range of blood-borne molecules to enter the pit; this “hemal compartment” (pale red) provides access to nanoparticles injected into the blood. The complex interlocking networks of hormonal and cytokine signaling, which maintain the integrity of the osteoclast-osteoblast team (the “bone modeling unit”) and which maintain the concomitant growth of the sprouting microvessel, are indicated by green lines (enhancement of function) and bright red lines (inhibition of function). The function of the microvessel in recruitment of osteoclasts and osteoblasts is indicated by the transendothelial migration of a monocytic cell at the middle left of the image. The cytosol of all cells is shown in pale blue, as in previous Figures. Fibrillar networks of the reticular system are omitted for clarity, as in previous Figures.

**Figure 6 ijms-22-08932-f006:**
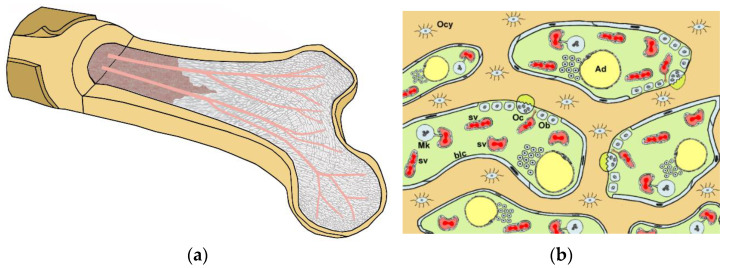
(**a**)**:** Cutaway schematic of a human long bone (femur). Cortical bone (beige) and cancellous bone (gray) are shown with the blood vessels (red) supplying the cancellous bone. The medullary cavity is brown. The cancellous bone consists of numerous trabeculae, amongst which the hematopoietic red blood marrow is located. (**b**): Close-up to show the detailed bone-blood relationships. The bone trabeculae (beige) and the “enclosed” osteocytes (Ocys) surround spaces containing the red bone marrow. The spaces are lined by flat bone-lining cells (blcs), and contain the bone marrow interstitial space (green). Within the interstitium, sinusoidal vessels (svs) contain blood; a small number of adipocytes (Ad, fat cells) are present. Megakaryocytes in the interstitium (Mk) protrude processes into the blood vessels, in order to release platelets into the blood. At the interface between interstitium and bone, resorption pits (“Howship’s lacunae”) are shown (out of scale), each with an osteoclast (Oc)-resorbing bone mineral (bright yellow) and accompanied by osteoblasts (Ob), which deposit osteoid as a bone replacement. The dense, complex reticular framework of the interstitium is omitted for clarity.

**Table 1 ijms-22-08932-t001:** Markers of bone formation, bone resorption and bone turnover that are potentially useful as targets in treating osteoporosis.

**Bone Formation Markers**					
**Origin**	**Type of Biomarker**		**Location**	**Clinical Utility**	**Assay**
Side-products of collagen synthesis	Procollagen type-I propeptides	PINP	blood	sensitive biomarker for bone formation rate	automated systems
		PICP		further studies required	radioimmunoassay
Enzymes from osteoblasts	ALP		blood	further studies required	automated systems
	BALP			further studies required	enzyme immunoassay, immunoradiometric assay
Matrix protein	OCN		blood	level of serum OCN can predict fracture risk; limited use as biomarker due to short half-life	ELISA
**Bone Resorption Markers**					
Enzymes from osteoclasts	CTSK		blood	potential marker for assessment of fracture risk	ELISA
	TRAP 5b			specific and sensitive marker for bone resorption	enzyme immunoassay, automated systems
Collagen degradation products	HYP		blood/urine	nonspecific biomarker for bone resorption; also present in skin and cartilage	
	HYL	GHYL	blood	derived from bone resorption only	HPLC
		GGHYL		derived from bone resorption and skin	
	PYD			specific marker for monitoring as it is mainly present in bone and dentin	enzyme immunoassay
	DPD		blood/urine	non-specific marker; also present in blood vessels, cartilage and ligaments	HPLC
	Telopeptides of type-I collagen	CTX-1	blood	high specificity and sensitivity but concentration is influenced by food intake	automated systems, ELISA
		CTX-MMP		not generally used as bone biomarker	
		NTX-1	blood/urine	high specificity and sensitivity; concentration is not influenced by food intake	ELISA
Non-collagenous proteins	OP		blood	potential use as biomarker for the assessment of osteoporosis treatment	ELISA
	BSP			in connection with OC and BALP, important predictive marker for bone resorption	radioimmunoassay
**Bone Turnover Regulators**					
Released by osteocytes	RANKL		blood	clinical relevance needs further investigation	ELISA
	OPG				
	DDK-1				
	Sclerostin				

**Legend to Table 1**: The abbreviations are according to their order of appearance in the table: PINP: Procollagen type-I N-terminal propeptide; PICP: Procollagen type-I C-terminal propeptide; ALP: Serum alkaline phosphatase; BALP: Bone-specific alkaline phosphatase; OCN, Osteocalcin or bone Gla protein; CTSK: Cathepsin K; TRAP 5b: Tartrate-resistant acid phosphatase 5b; HYP: Hydroxyproline; HYL: Hydroxylysine; GHYL: Galactosyl-hydroxylysine; GGHYL: Glucosyl-galactosyl-hydroxylysine; PYD: Pyridinoline; DPD: Deoxypyridinoline; CTX-1: Carboxy terminal crosslinked telopeptide; CTX-MMP: Carboxy terminal crosslinked telopeptide matrix metalloproteinases; NTX-1: Amino terminal crosslinked telopeptide; OP: Osteopontin; BSP: Bone sialoprotein; RANKL: Receptor activator of NF-κB ligand; OPG: Osteoprotegerin; DDK-1: Dickkopf-1.

**Table 2 ijms-22-08932-t002:** Overview of recent developments of nanoparticles designed for treating osteoporosis.

**Organic Nanoparticles**				
**Type of Nanoparticle**	**Target**	**Therapeutic**	**Ex Vivo/In Vivo**	**Reference**
Chitosan NPs	BMP-2		in vivo rat model	[121]
Polyurethane NPs	anti-miR214		in vivo mouse model	[122]
Dendrimer based NPs	C11 peptide CH6 aptamer		in vivo	[123]
HPMA copolymers	D-Asp_8_		in vivo rat model	[124]
Poly(L-lactide-co-glycolide) NPs (PLGA)		risedronate	in vivo rat model	[125]
Lipid NPs		simvastatin	ex vivo	[126]
	antagomir-148a		in vivo mouse model	[127]
**Inorganic Nanoparticles**				
Mesoporous silica NPs			ex vivo	[128]
Titanium nanotubes		alendronate	in vivo rabbit model	[129]
Gold NPs		alendronate	in vivo mouse model	[130]
Nanodiamonds		alendronate	in vivo mouse model	[131]
Calciumphosphate NPs	long-chain microRNA-34a conjugate		ex vivo	[132]
Hydroxyapatite		risedronate	in vivo rat model	[133]
	calcitonin		in vivo rabbit model	[134]

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
