# Peer review of "Targeting Drug Delivery in the Elderly: Are Nanoparticles an Option for Treating Osteoporosis?"

_ijms, 2021, doi:10.3390/ijms22168932_

Round 1

Reviewer 1 Report

This manuscript reviews the use of nanomedicine for osteoporosis treatment, explaining the potential targets, current situation and future perspectives. In general, it is well written and well referenced. It deserves to be published in IJMS. Here there are some comments: 

Please, identify the diseases. Cancer, infectious diseases….  Maybe a table including type of nanocarrier, the approved formulation (e.g., by FDA and EMA) is useful.

-  Introduction: I suggest including more information about osteoporosis, its treatment strategies, and their problems.

- Did the authors plan to explain section 7 after section 8 and 9?

- I suggest changing the title of section 10, something more impersonal. 

Reviewer 2 Report

It was a good review about the possibility of utilizing targeted drug delivery for the treatment of osteoporosis. Here are some minor comments about this study that should be considered before publication:

  • Please improve the quality of the abstract.
  • Please improve the quality of section "2. Targeting to Tissues in vivo"
  • Please add more samples for organic nanoparticles in table1.
  • Please use more updated references.

Reviewer 3 Report

It's interesting and well written (mini)review.

I have just some minor comments.

The Table 2 doesn't correcpond to article title. Please  change the title or please cosider changing the table 2 content  or moving it to the supplementory material ( which would be however unfavorable to the manuscript).

I know that EPR abbrevation is explained in the abstract and in Abbrevation list at the end of the manuscript, but please consider explain it when first used in the main text (L110).

Please consider changing "Howship lacuna" to "Howship's lacuna". Also please unify: space of Disse (L219) or Disse space (L514). Simirarly, please unify the numenclature (uppercase or lowecase) in drugs names (like Denosumab (L434) or denosumab (L567)).

L239 "recycling of bone" - wording

Table 1 - I believe that there are also other more recent (the last 4 years) studies like doi.org/10.1016/j.nano.2020.102153 available.
